# Dopamine-dependent scaling of subthalamic gamma bursts with movement velocity in patients with Parkinson's disease

Roxanne Lofredi[1]\*, Wolf-Julian Neumann[1], Antje Bock[1], Andreas Horn[1], Julius Huebl[1], Sandy Siegert[1], Gerd-Helge Schneider[2], Joachim K Krauss[3], Andrea A Kühn[1]\*

[1]Department of Neurology, Campus Mitte, Charité – Universitätsmedizin Berlin, Berlin, Germany; [2]Department of Neurosurgery, Campus Mitte, Charité – Universitätsmedizin Berlin, Berlin, Germany; [3]Department of Neurosurgery, Medical School Hannover, MHH, Hannover, Germany

**Abstract** Gamma synchronization increases during movement and scales with kinematic parameters. Here, disease-specific characteristics of this synchronization and the dopamine-dependence of its scaling in Parkinson's disease are investigated. In 16 patients undergoing deep brain stimulation surgery, movements of different velocities revealed that subthalamic gamma power peaked in the sensorimotor part of the subthalamic nucleus, correlated positively with maximal velocity and negatively with symptom severity. These effects relied on movement-related bursts of transient synchrony in the gamma band. The gamma burst rate highly correlated with averaged power, increased gradually with larger movements and correlated with symptom severity. In the dopamine-depleted state, gamma power and burst rate significantly decreased, particularly when peak velocity was slower than ON medication. Burst amplitude and duration were unaffected by the medication state. We propose that insufficient recruitment of fast gamma bursts during movement may underlie bradykinesia as one of the cardinal symptoms in Parkinson's disease.
DOI: https://doi.org/10.7554/eLife.31895.001

**\*For correspondence:**
roxanne.lofredi@charite.de (RL);
andrea.kuehn@charite.de (AAKü)

**Competing interests:** The authors declare that no competing interests exist.

## Introduction

Oscillatory brain rhythms are thought to contribute to the processing and long-range transmission of information as encoded in single-neuron activity through coherence of neuronal assemblies (*Fries, 2015*). Pathological oscillatory patterns could be related to disease-specific symptoms in Parkinson's disease using invasive recordings in patients undergoing deep brain stimulation surgery for severe movement disorders (*Deuschl et al., 2006*; *Dams et al., 2016*). Most strikingly, an elevated synchronization in the subthalamic beta band activity is found in Parkinson's disease that correlates with symptom severity in the dopamine-depleted state (*Williams et al., 2003*; *Neumann et al., 2016b*; *Kühn and Volkmann, 2017*) and is attenuated by dopaminergic replacement therapy (*Kühn et al., 2006, 2009*) as well as deep brain stimulation (*Eusebio et al., 2011*; *de Hemptinne et al., 2015*; *Oswal et al., 2016*). In contrast to pathological activity at rest, less is known about the functional role of physiological movement-related changes of oscillatory patterns in the cortico-basal ganglia motor loop. In the human motor system, ongoing movement is accompanied by increases in cortical (*Crone et al., 1998*; *Cheyne, 2013*) and subcortical gamma (30–90 Hz) oscillations (*Androulidakis et al., 2007*; *Kempf et al., 2009*; *Brücke et al., 2012, 2013*) and strengthening of

**eLife digest** Parkinson's disease is a disorder of the nervous system that affects more than 1% of people over the age of 60. Symptoms include uncontrollable shaking or tremor, and difficulty with large or fast movements. These symptoms occur when neurons that produce the chemical dopamine die. The loss of dopamine disrupts the activity of structures deep within the brain called the basal ganglia, which normally control movement.

Some patients with Parkinson's disease benefit from a treatment known as deep brain stimulation. Electrodes are lowered into the brain to stimulate part of the basal ganglia called the subthalamic nucleus. But we can also use these electrodes to record the activity of neurons. Doing so reveals that during movement, neurons in the subthalamic nucleus coordinate their firing at a frequency of 40 to 90 hertz. This is known as gamma synchronization.

Lofredi et al. now reveal that patients with Parkinson's disease, who do not take any medication, show reduced gamma synchronization. The greater the loss of synchronization, the more slowly patients move. Gamma synchronization does not occur continuously during a movement, but instead occurs in brief bursts. Patients with Parkinson's disease show a reduction in the number of bursts, but not in their duration or intensity.

Measuring bursts of gamma synchronization may help reveal what is happening inside the brain of a patient with Parkinson's disease in real time. This could lead to improvements in deep brain stimulation therapy. At present, electrodes stimulate the basal ganglia continuously, but this can lead to side-effects. In the future, it may be possible to apply stimulation only when there is too little synchrony. This could reduce side effects and make the treatment more effective.

DOI: https://doi.org/10.7554/eLife.31895.002

coupling in that frequency range (*Litvak et al., 2012*). Cortical and subcortical movement-related gamma band synchronization seems a general and physiological phenomenon irrespective of the underlying disease. The power increase in high frequencies most likely relies on mechanistically different phenomena with a primarily oscillatory component in the frequency range 30–90 Hz and overlapping asynchronous spiking activity in frequencies >100 Hz (*Fries et al., 2007*; *Manning et al., 2009*; *Buzsáki and Wang, 2012*). Previous studies have shown that narrowband increases in gamma oscillations with a center frequency between 30–90 Hz and ~20 Hz width were of functional significance in encoding of movement parameters (*Jenkinson et al., 2012*; *Joundi et al., 2012*; *Tan et al., 2015*). In line with this, human and animal studies have implicated that the basal ganglia play a major role in scaling of movement (*Desmurget et al., 2003*; *Kravitz et al., 2010*; *Grafton and Tunik, 2011*; *Brücke et al., 2012*; *Joundi et al., 2012*; *Oldenburg and Sabatini, 2015*; *Tan et al., 2015*). The loss of their physiological activity may lead to hypo- or hyperkinetic movement disorders. Whether the encoding of movement scaling through subcortical oscillatory activity in the gamma band is dopamine-dependent, thus impaired in PD patients after withdrawal of dopaminergic medication, and correlates with movement slowing in the OFF dopaminergic state of the same patient, has to our knowledge never been investigated. Moreover, recent studies have highlighted that both task-specific and resting state oscillatory synchronization do not reflect a continuous activity but rather occur in bursts. While this has been primarily shown for beta activity in the basal ganglia (*Feingold et al., 2015*), intracerebral recordings of cortical areas highlighted an encoding capacity through variations of gamma burst density in memory processing (*Kucewicz et al., 2017*; *Lundqvist et al., 2016*). In Parkinson's disease, there is emerging evidence for a disadvantageous distribution toward long beta bursts (*Tinkhauser et al., 2017*). In the gamma frequency range, neither the phenomenon of transient synchrony in motor processing nor how it may be influenced by dopamine signaling has so far been investigated. Here, we aimed at elucidating the patho-physiological role of velocity-related subthalamic gamma oscillations and their modulation by dopamine in patients with Parkinson's disease. We hypothesized that movement-induced gamma oscillations that have been shown to drive the motor cortex (*Litvak et al., 2012*) are gradually scaled by kinematic velocity as seen in the upstream globus pallidus internus (*Brücke et al., 2012*), the main projection-target of the subthalamic nucleus and major output nucleus of the basal ganglia. A coding deficiency in modulating subcortical gamma signaling through reduced burst rate in the hypodopaminergic

state may contribute to the underlying pathomechanism of bradykinesia, one of the cardinal symptoms in Parkinson's disease.

## Results

### Behavioral results

The mean reaction time (±SEM) of the 16 patients was 651 ± 176 ms across all trials, which was similar (p=0.3, permutation test) to the reaction time in the subgroup of seven patients that additionally performed the task after withdrawal of the dopaminergic therapy (reaction time-OFF: 622 ± 43 ms; small: 613 ± 43 ms, medium: 617 ± 39 ms, large: 637 ± 47 ms). Between movement conditions, reaction time was slightly but significantly shorter in the small compared to the medium and large condition (small: 621 ± 23 ms, medium: 651 ± 23 ms, large: 648 ± 23 ms, small-medium: p=0.02, small-large: p=0.002) in the patient cohort ON medication (n = 16) and in the small compared to the large condition in the seven patients OFF medication (small – large: p=0.03). No difference between conditions was seen in those seven patients ON medication. In both medication states, there was a significant stepwise increase of both movement amplitude and movement velocity (small <medium < large, p<0.001) between the three movement conditions in the time window from −183 ms to 670 ms around maximal velocity (small-medium: −172 to 800 ms, medium-large: −150 to 670 ms, small-large: −190 to 800 ms around maximal velocity, p<0.05, FDR-corrected), see *Figure 1*. Patients were not significantly slower in the OFF compared to the ON state (n = 7, p=0.1) when averaged across all conditions but showed a significant decrease of movement velocity when performing the large condition (p=0.05) as shown in *Figure 1*. This indicates that patients in the OFF-dopaminergic state were primarily impaired in the execution of large and fast movements, while smaller and respective slower movements were less affected without dopaminergic replacement therapy.

### General features of relative subthalamic power changes with movement

Time frequency representations averaged across all bipolar recordings and subsequently across patients revealed a specific movement-related pattern of synchronization and desynchronization that was significantly different from baseline activity (p<0.05, FDR-corrected) in the subthalamic nucleus both contralateral and ipsilateral to the moved side (*Figure 2*, for the ipsilateral side, see *Figure 2—figure supplement 2*). In all conditions, movement onset was preceded and accompanied by an ERS in the theta frequency range (2 to 8 Hz) and an ERD in the beta band (13 to 30 Hz). The theta ERS started ~1.2 s before movement onset and persisted up to ~2.7 s after movement onset. The beta ERD started later (~0.4 s before movement onset) and ended about ~1.7 s after movement onset. In addition, a pronounced synchronization in the gamma frequency range, spreading from 35 to 100 Hz, started shortly before movement onset at about ~0.3 s and persisted up to ~1.4 s for the smallest and ~3 s for the largest movement. This gamma synchronization during movement seems to segue into a post-movement beta ERS. However, when interpreting the results it should be kept in mind that the cycle number of the wavelet transform may affect the exact starting and ending time points of the desynchronization and synchronization phenomena.

### Parametric modulation of event-related oscillatory patterns

Gamma power around maximal velocity was first averaged across all contacts per electrode, second across hemispheres of one patient for statistical testing and then averaged across patients for visualization. Permutation testing of averaged time-frequency bands revealed a stepwise increase of gamma power with movement velocity in both hemispheres (contralateral: small: 5.7%, medium: 9.4%, large: 11.6%, small vs large, p<0.001, small vs medium, p<0.001, medium vs large, p=0.02; ipsilateral: small: 4.1%, medium: 7.2%, large: 9.4%, small vs large, p=0.004, small versus medium, p=0.004, medium versus large, p=0.01; FDR-corrected), see *Figure 2*. No such modulation was seen in the theta or beta band. Direct comparison of relative power changes in all time-frequency bins across movement conditions (large, medium or small movement) confirmed a parametric movement-related modulation of oscillatory activity in the contralateral hemisphere that was essentially restricted to the gamma band, ranging from 32 to 100 Hz in a period of up to 2.7 s after movement onset. The time-frequency clusters that displayed a significant change in mean oscillatory amplitude

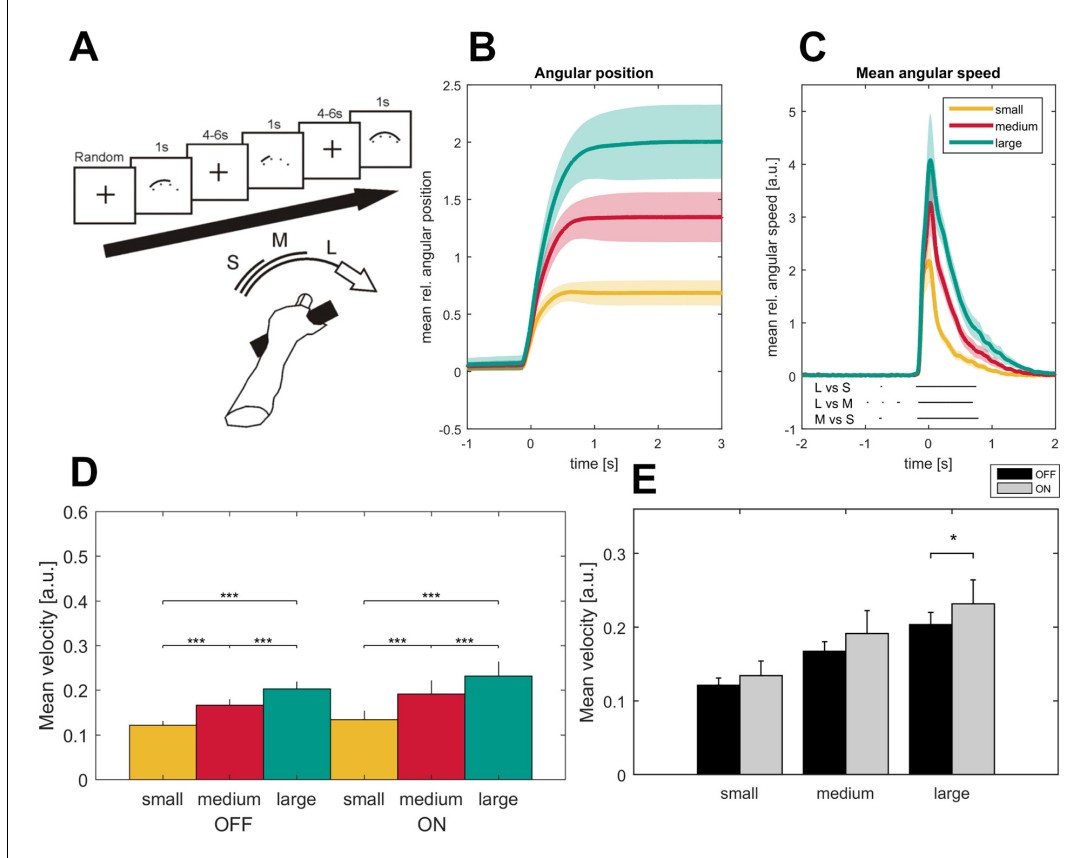

**Figure 1.** Task description and movement traces.  Patients were asked to perform cued forearm pronation movements of three different sizes (A) after presentation of a baseline fixation cross that resulted in hand movements of three different amplitudes (B; small - yellow, medium - red, large - green) and movement velocities (C) aligned on movement onset. Shaded areas indicate the standard error of the mean. Black lines below the movement velocities indicate statistical significance (p<0.05, FDR corrected). Movement velocity shows a stepwise increase toward the large movement condition in the ON- and OFF- medication state (D). Patients in the dopaminergic OFF state (n = 7, sides averaged for each patient) performed the task significantly slower only in the large movement condition (E).

DOI: https://doi.org/10.7554/eLife.31895.003

when compared to one of the other movement conditions are shown in *Figure 2*, right column (p<0.05, permutation test, FDR-corrected).

## Movement-related activation pattern in the ipsi- and contralateral subthalamic nucleus

Movement-related changes in oscillatory activity were seen in both the ipsi- and the contralateral subthalamic nucleus to the moved side. To test for differences between ipsi- and contralateral hemispheres we conducted permutation tests separately for the previously defined time-frequency ranges across conditions and separately for each condition. Only the synchronization in the gamma band showed a trend toward a more pronounced gamma ERS in the subthalamic nucleus contralateral to the moved limb (theta band: p=0.4, beta band: p=0.8, gamma band: p=0.08), as seen in *Figure 2*. Interestingly, the difference between gamma ERS of the ipsi- and contralateral subthalamic nucleus becomes significant when comparing the movement conditions separately. While large and medium movements entrain a significantly stronger gamma synchronization in the subthalamic nucleus contralateral rather than ipsilateral to the moved side, small movements go along with a similar oscillatory activity in both subthalamic nuclei (small: p=0.9, medium: p=0.01, large: p=0.03, permutation test) as shown in *Figure 2E*. In contrast, mean relative power changes were similar in both hemispheres for all conditions in the theta band (small: p=0.8, medium: p=0.4, large: p=0.5, permutation test) and the beta band (small: p=1, medium: p=1, large: p=0.5, permutation test).

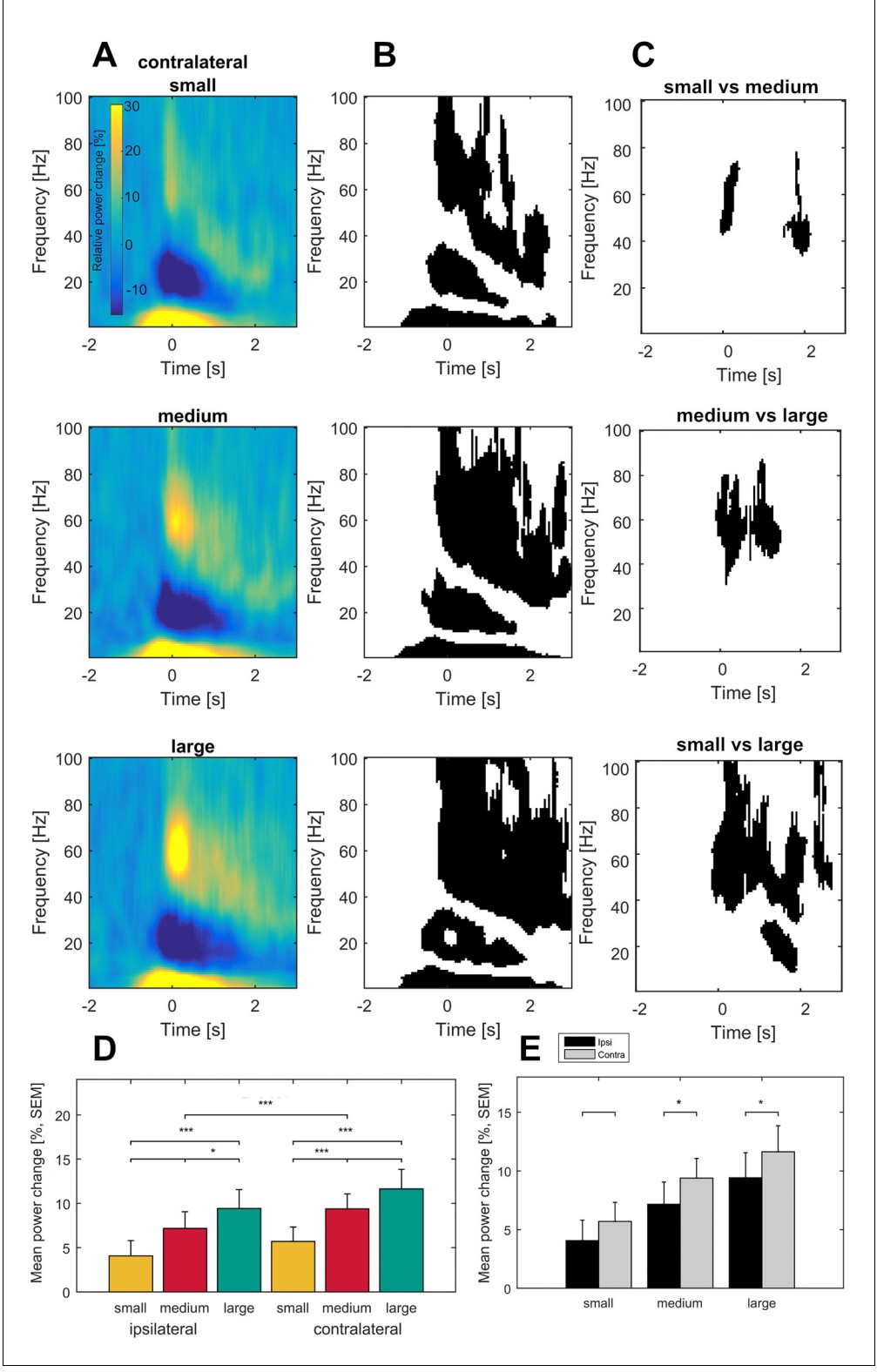

**Figure 2.** Grand average of subthalamic oscillatory activity aligned to movement onset in the dopaminergic ON state. Baseline corrected time-frequency representations were averaged across contact pairs of one electrode, hemisphere in relation to the moved hands of each patient and across patients (**A**). Note the scaling of gamma amplitude across movement conditions (small – top, medium – middle, large - bottom). Statistical analysis of time-frequency representations across patients revealed significant movement-related modulation from baseline in the

*Figure 2 continued on next page*

*Figure 2 continued*

theta, beta and gamma band in all conditions (B, p<0.05; FDR corrected). (**C**) Significantly different time-frequency bins were predominantly in the gamma band and highest in the comparison of small and large movements (lower row) in the contralateral hemisphere. All p-values<0.05 after FDR correction for multiple comparisons. (**D**) Across patient averages (n = 16 ON medication) revealed a stepwise increase in gamma power (40–90 Hz) toward the large movement condition (small – yellow, medium – red, large – green) both ipsi- and contralateral to the moved side (ipsilateral – left, contralateral – right). In the contralateral hemisphere, stronger gamma synchronization was seen in the medium and large movement conditions when compared to the ipsilateral hemisphere (**E**; grey–contralateral, black – ipsilateral). No significant modulation by movement condition and hemisphere was seen in the theta (2–8 Hz) and beta bands (13–30 Hz) (not shown).

DOI: https://doi.org/10.7554/eLife.31895.004

The following figure supplements are available for figure 2:

**Figure supplement 1.** Averaged power spectra at rest and movement in an exemplary patient and across all patients.

DOI: https://doi.org/10.7554/eLife.31895.005

**Figure supplement 2.** Grand average of subthalamic oscillatory activity in the ipsilateral hemisphere aligned to movement onset in the dopaminergic ON state.

DOI: https://doi.org/10.7554/eLife.31895.006

**Figure supplement 3.** Averaged beta power (13–30 Hz) over trials ON and OFF medication.

DOI: https://doi.org/10.7554/eLife.31895.007

## Correlation of oscillatory activity with behavior and clinical state

As seen in the behavioral data, increasing movement amplitude correlates with a significant increase in movement velocity. To further disentangle the influence of movement velocity and amplitude on gamma synchronization, gamma-band activity contralateral to the moved side was averaged across the three movement conditions and aligned to the time points movement onset, maximal velocity and maximal amplitude. Mean gamma power (0–0.5 s following the respective time point) was compared between the three alignment points. This revealed a significantly stronger synchronization in the gamma band when aligned on maximal velocity (14.5%) compared to maximal amplitude (6.9%, p<0.001) and movement onset (12.3%, p=0.02), see *Figure 3*. In all patients, there was a positive correlation between contralateral gamma band power (40–90 Hz, ±0.5 s around maximal velocity) and the maximally reached velocity on a single-trial level that was significant in 8/16 patients. We further investigated the correlation for the entire spectrum of oscillatory activity and maximal velocity on a single-trial level separately for the hemispheres ipsi- and contralateral to the moved hand. When averaging the resulting correlation plots across all patients, a weak but significant positive correlation between relative power changes in the subthalamic nucleus contralateral to the moved hand and maximal velocity within a trial extended from 24 to 95 Hz and −400 to +600 ms around maximal velocity (p<0.05, FDR-corrected), as seen in *Figure 3*. The significant cluster was prominent in the gamma band and reflects a finely tuned scaling of velocity-related power changes on a single-trial level. Importantly, the subthalamic nucleus ipsilateral to the movement side lacked correlation between velocity and relative power changes. The significant correlation between relative power changes and behavior on a single-trial level was specific for velocity. Other tested parameters such as reaction time, amplitude and accuracy did not significantly correlate across all patients on a single-trial level. To control for a potential fatigue effect leading to decreased peak velocity over trials or a subsequent compensatory mechanism in the gamma band, we investigated a potential systematic change of movement velocity or gamma band activity over the course of the experiment. There was no significant correlation between either variables over trials (Spearman's Rho = −0.09, p=0.3) making a systematic fatigue effect unlikely. Correlation between motor impairment as assessed by the UPDRS-ON score and averaged relative power changes around maximal velocity showed a negative correlation between both mean theta (Spearman's ρ = −0.73, p=0.006) and mean gamma power (Spearman's ρ = −0.55, p=0.038) and the individual UPDRS-ON score as shown in *Figure 4*. In contrast, averaged beta desynchronization during movement did not correlate with the symptom severity ON medication.

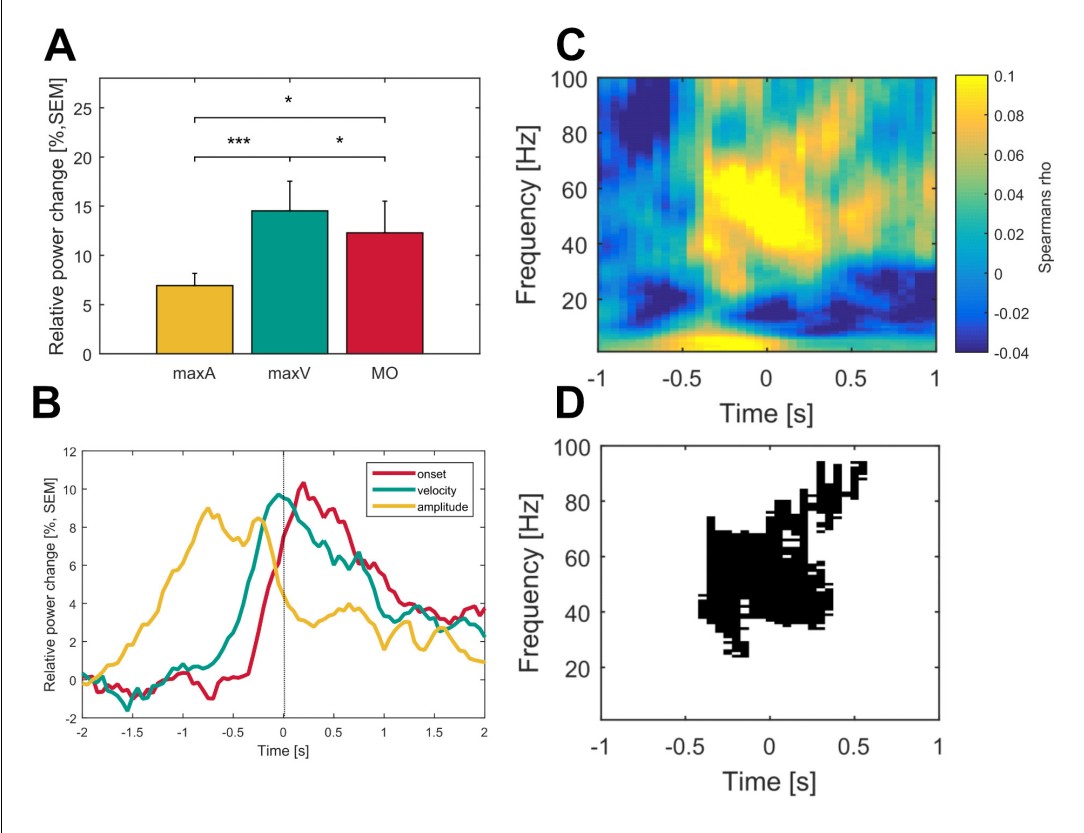

**Figure 3.** Gamma activity is strongest during maximum movement velocity and directly correlates with movement velocity on a single-trial level across patients. Averaged gamma band activity was aligned to different time points of the movement execution (A; onset – red, velocity – green, amplitude – yellow) to allow comparison of a temporal focus in gamma synchronization. (B) Gamma activity was significantly higher aligned to the time point of maximum velocity (green) when compared to an alignment on maximum amplitude (yellow, p<0.001) and movement onset (red, p=0.02). (C) Single-trial Spearman's correlations with movement velocity across all time frequency bins aligned to maximum movement velocity (time 0 s) were conducted in all patients. (D) Robust positive correlations were spectrally and spatially distinct to the gamma band (p<0.05, FDR corrected) of the contralateral hemisphere. No significant time frequency bins were revealed for the ipsilateral hemisphere (data not shown).

DOI: https://doi.org/10.7554/eLife.31895.008

### Dopamine-dependence of movement-related modulation in oscillatory patterns

Since we observed movement-related fine tuning of gamma-band synchronization mainly in the contralateral side, we focused our subgroup analysis of dopamine-related changes to the hemisphere contralateral to the moved hand. In both dopaminergic states, movement-related changes in oscillatory activity as seen when averaged across patients showed a similar general pattern with movement-related beta ERD and theta and gamma ERS (*Figure 5A,B*). However, mean gamma synchronization during movement was significantly less pronounced in patients OFF medication (relative power change ON = 15%, OFF = 9%, p<0.001, permutation test, *Figure 5C*). When comparing movement conditions separately this lack in gamma power increase gradually enlarged toward the large movement condition (difference ON - OFF: small = 7%, medium = 13%, large = 20%, *Figure 5D*). Similarly, on a behavioral level subjects OFF medication showed significant movement slowing during the large movement condition (p=0.05; *Figure 1*). Hence, parkinsonian motor impairment may be related to disturbed modulation capacity in gamma power scaling. Consequently, the parametric scaling of contralateral gamma synchronization with movement condition was lost OFF medication (small vs medium, p=0.09; medium vs large, p=0.5) with only a significant difference between small and large movements (p=0.018; *Figure 5C*). Theta ERS remained similar in both conditions. Relative changes in beta desynchronization differed between time points. While beta ERD was significantly more pronounced in the ON medication state when averaged around

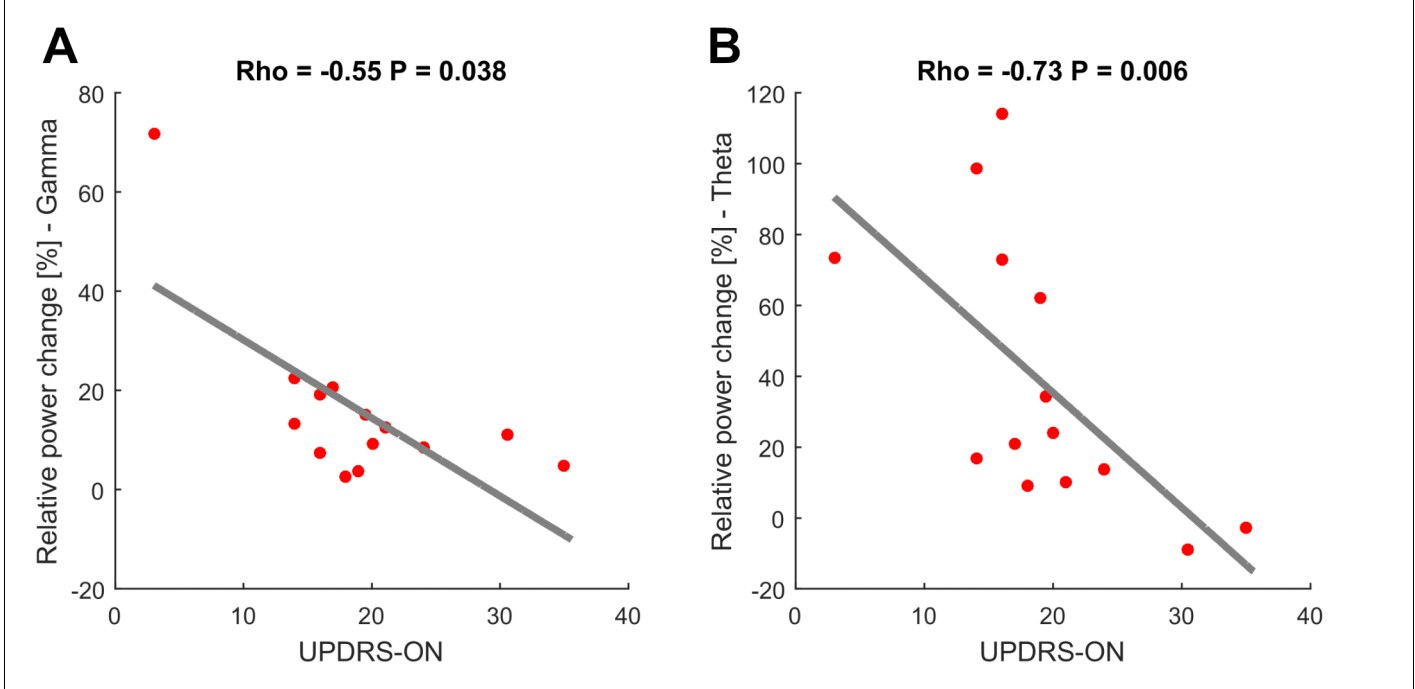

**Figure 4.** Movement-related theta and gamma oscillations are correlated with parkinsonian symptom severity in the dopaminergic ON state. To investigate a potential association of movement-related activity, Spearman's correlations were conducted between averaged theta (2–8 Hz), beta (13–30 Hz) and gamma (40–90 Hz) amplitudes with concurrent motor impairment as assessed by UPDRS-III (available in 14/16 patients). Significant negative correlations were found for the theta (A; Spearman's ρ = −0.073, p<0.01) and gamma (C; Spearman's ρ = −0.55, p=0.038) amplitudes.

DOI: https://doi.org/10.7554/eLife.31895.009

movement onset (ON: −29.37 ± 19.27%, OFF: −16.36 ± 16.01%, p=0.037), this difference did not reach significance during maximal velocity (ON: −29.5 ± 19.5%, OFF: −22.3 ± 19.3%, p=0.062), see *Figure 2—figure supplement 3*.

## Properties of gamma bursts during movement in both dopaminergic states

On average, 42.6 ± 3.5 gamma bursts with a duration of 27.1 ± 1.5 ms occurred in each trial. Gamma peak frequency was 69.4 ± 1.17 Hz (range 62–81 Hz), suggesting that one gamma burst included in average 1.9 cycles of gamma oscillations. In the ON medication state, all gamma burst features increased significantly with movement when compared to baseline (26 hemispheres of 16 patients), see *Figure 6B*. Across patients, burst rate (bursts per 0.5 s) increased from 2.8 ± 0.25 to 3.4 ± 0.25 bursts during movement (p<0.001), burst amplitude by 51 ± 0.11% (p<0.001) and burst duration from 26.9 ± 2 ms to 29 ± 1.9 ms (p<0.001), see *Figure 6B*. Relative changes in burst rate during movement correlated significantly higher with gamma power changes (Spearman's ρ = 0.8) than burst duration (Spearman's ρ = 0.39, Burst rate vs amplitude: p=0.009) and burst amplitude (Spearman's ρ = 0.42, Burst rate vs duration: p=0.05), see *Figure 6C*. Additionally, relative changes in gamma burst rate during movement revealed a scaled increase toward the large movement condition (large: 48.2 ± 6.5%, medium: 36.7 ± 5%, small: 29.3 ± 3.5%, L vs M: p=0.01, L vs S: p=0.003, M vs S: p=0.09), see *Figure 6D*. Averaged relative changes in burst rate during movement correlated negatively with the UPDRS-ON score that was available in 14 patients (normally distributed, Pearson's r = −0.6, p=0.008), see *Figure 6E*. In the dopamine-depleted state, gamma burst rate during movement showed a significant decrease compared to the ON medication state (ON: 55 ± 4%, OFF: 24 ± 4%, p<0.001) while increase in burst amplitude (ON: 58.8 ± 16%; OFF: 39.6 ± 2.4%, p=0.4) and duration remained similar (ON: 16.65 ± 0.04 ms; OFF: 10.9 ± 0.01 ms, p=0.2), in 10 contralateral hemispheres of seven patients (*Figure 6F*).

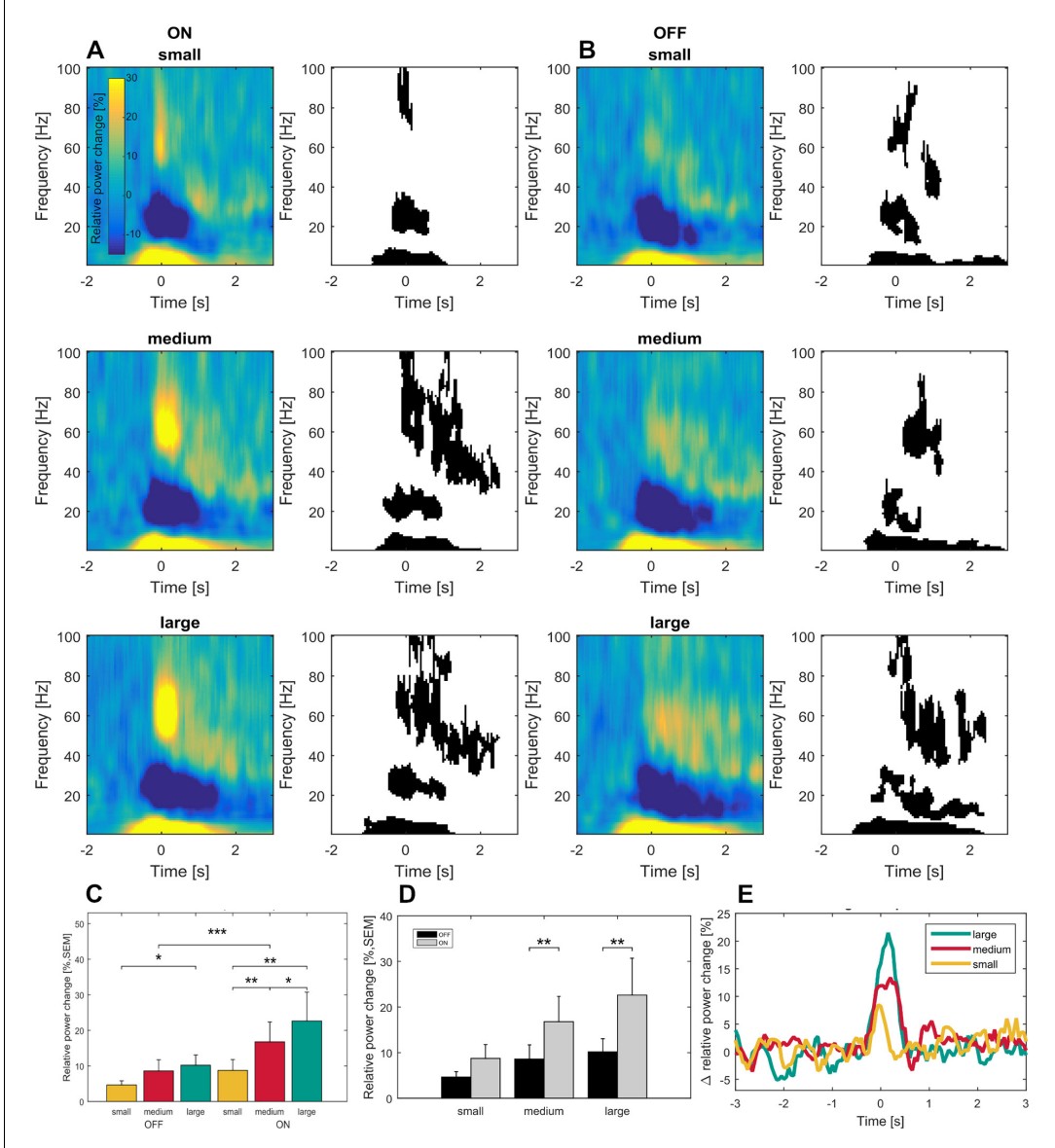

**Figure 5.** Time frequency representations and dopamine-dependent-scaling of gamma activity in the subgroup of seven patients ON and OFF medication. Significant movement-related modulation of the contralateral baseline corrected time-frequency representation was found for theta, beta and gamma frequency bands (ON: A- right panel, OFF: D – right panel; p<0.05, FDR corrected) in the dopaminergic ON (A) and OFF (B) state in a subgroup of seven patients. Scaling of movement to gamma oscillations was present only in the dopaminergic ON, but not in the OFF state (C). Gamma band activity difference between ON and OFF states were scaled to movement velocity (D) and significantly different only in the medium and large movement conditions in the subgroup of seven patients with OFF recordings. No significant difference was found for the theta and beta bands (not shown). An increasing difference in gamma power between ON and OFF state toward the large movement condition is shown in (E). Across conditions, grand average gamma activity was significantly higher in the dopaminergic ON state, when compared to the OFF state (C; p<0.001).
DOI: https://doi.org/10.7554/eLife.31895.010

## Spatial distribution of gamma oscillatory reactivity across patients

Spatial interpolation of gamma power values mapped to contact pair location in MNI space showed a clear peak in gamma power in the dorsolateral portion of the subthalamic nucleus overlapping the subthalamic motor area (*Figure 7*). Correlations between averaged gamma power and each coordinate axis revealed significantly increased gamma synchronization for the lateral contact pair position (X-axis) and trends for more anterior (Y-axis) and dorsal (Z-axis) contact pair positions (Spearman's

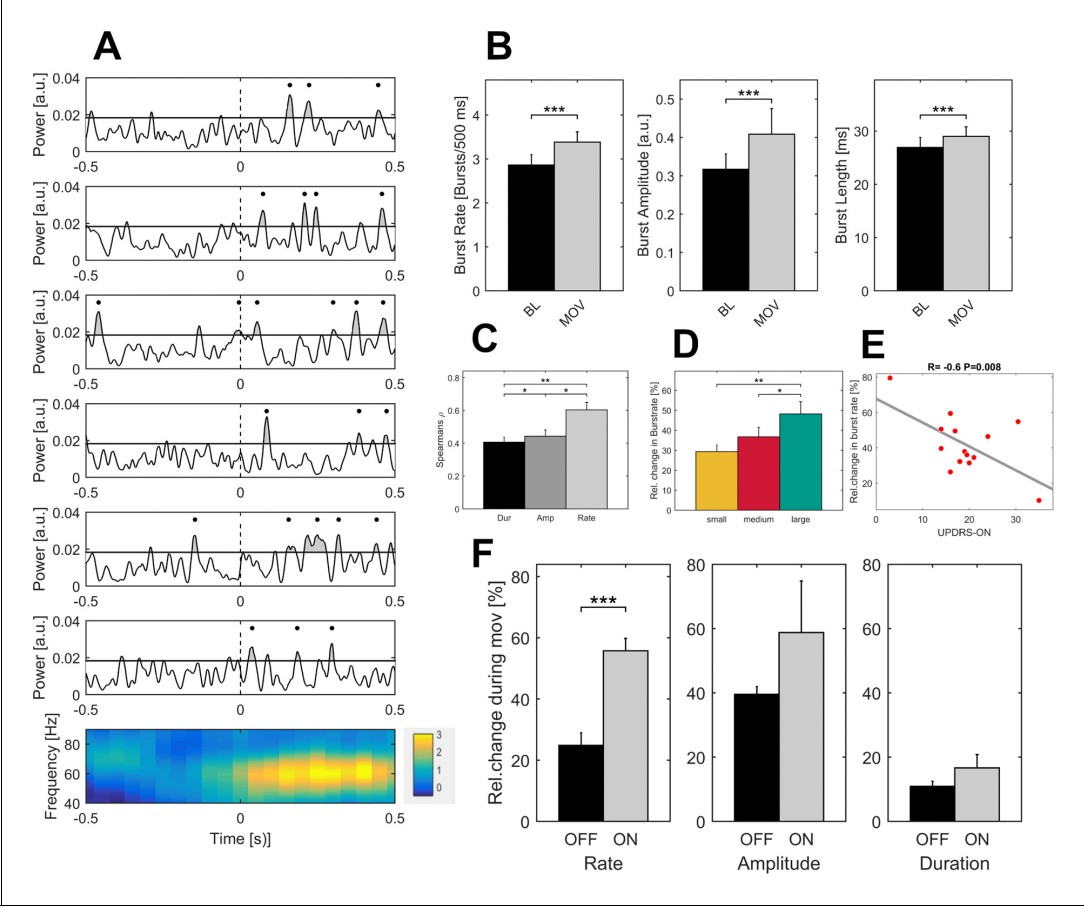

**Figure 6.** Properties of gamma bursts scale during movement and indicate both the dopaminergic and the clinical state. (A) The raw LFP signal was filtered around the gamma range (40–90 Hz) and for all contact pairs of one electrode, the same threshold was set at the 75th percentile of the gamma amplitude of the entire recording. A burst was considered when the duration of threshold crossing (black line) was at least one gamma cycle long (black dot). Shown are six trials of the large movement condition from the same patient (Subject 2, right hemisphere, contact pair 1–2) and the averaged time-frequency plot across large condition trials for this patient (bottom row). After movement onset (dashed line), the rate of transient synchrony in gamma bursts increases while the overall level of gamma power appears unchanged. Gamma bursts occur at different time points within each trial leading to a seemingly continuous gamma synchronization when averaging across trials (bottom row). (B) When averaged across patients (n = 16, ON-state), gamma bursts show increased rate, amplitude and duration during movement (grey bar – MOV), compared to baseline (black bar – BL). (C) Averaged correlations of changes in gamma-power and –burst properties for each patient showed significantly higher correlation with increases in burst rate (light grey - Rate) than burst amplitude (grey - Amp) or duration (black – Dur). (D) Relative changes of burst rate show a stepwise increase toward the large movement condition. (E) Burst rate increase correlates with clinical state. UPDRS-ON scores were available in 14 patients. Across these patients, a significant negative correlation (Pearson's r = −0.6, p=0.008) was seen between increases in burst rate during movement and the UPDRS-III motor score. (F) When comparing averaged increases in burst rate, amplitude and duration ON and OFF medication (n = 7), only burst rate showed a significant decrease, while movement-related increases in burst duration and amplitude seemed not associated with the dopaminergic state. * indicate p-values<0.05; **p<0.01; ***p<0.001.

DOI: https://doi.org/10.7554/eLife.31895.011

ρ = 0.37/0.21/0.20, p=0.002/0.06/0.06 for X/Y/Z axis positions, permutation tests, FDR corrected for multiple comparisons).

## Discussion

Our results provide first evidence for dopamine-dependent and finely tuned scaling of movement-related gamma synchronization that occurs in transient bursts and is spatially confined to the dorso-lateral part of the subthalamic nucleus in patients with Parkinson's disease. While movement was accompanied by a broadband increase in high frequency activity (40–150 Hz), the parametric modulation with maximal movement velocity on a single-trial level was spectrally restricted to a narrow

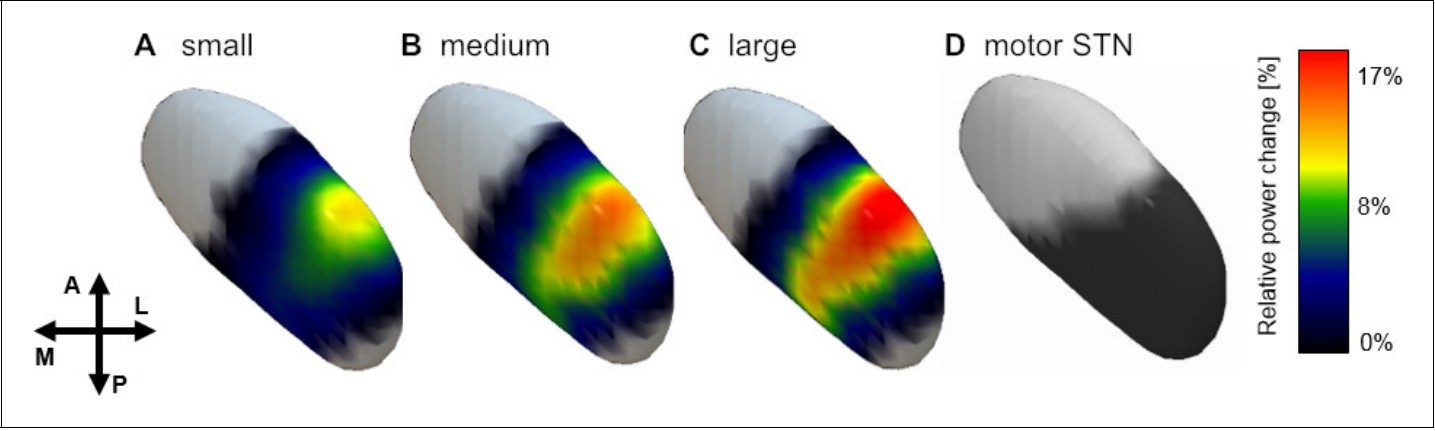

**Figure 7.** Spatial distribution of gamma power across conditions. Averaged gamma power for small (A), medium (B) and large (C) conditions was spatially interpolated between all contact pairs (left electrodes flipped to right hemisphere) and mapped to the subthalamic surface in standard MNI space. The resulting map revealed a clear peak in the dorsolateral portion of the subthalamic nucleus, which overlapped with the subthalamic motor segment (*Ewert et al., 2017*) shown in (D). Note the parametric peak size increase with larger movement amplitude.
DOI: https://doi.org/10.7554/eLife.31895.012

The following figure supplement is available for figure 7:

**Figure supplement 1.** Three-dimensional electrode reconstruction.
DOI: https://doi.org/10.7554/eLife.31895.013

gamma band (40–90 Hz) in the contralateral hemisphere. Therefore, we hypothesize that two co-occurring phenomena take place in the subthalamic nucleus during movement: a narrowband gamma oscillation that scales with motor output and an asynchronous broadband activity that may reflect local spiking activity without a direct behavioral correlate. The synchronization of narrowband gamma power during movement was not continuous but occurred in bursts. The increase of burst rate correlated with higher average of gamma power during movement. Both increase in movement-related gamma power and gamma burst rate correlated negatively with parkinsonian motor impairment, as assessed by the UPDRS-III score. In the OFF medication state, the parametric increase in gamma power with faster movements was lost and gamma burst rates significantly decreased in parallel with apparent movement slowing when patients had to perform the large movement condition. The evanescence of gradual gamma recruitment in rapid consecutive bursts during movement might therefore play a crucial role in the appearance of bradykinesia as one of the cardinal symptoms in Parkinson's disease. Movement-related changes in the theta and beta band showed no velocity-related modulation and were similarly pronounced bilaterally. The relative power change around maximal velocity was not dopamine dependent. However, beta desynchronization was significantly stronger in the ON vs OFF condition, when trials were aligned to movement onset, which may indicate an association of beta decrease with movement initiation. Movement-related averaged theta band synchronization correlated positively with the individual UPDRS score ON medication, highlighting its potential prokinetic role in Parkinson's disease (*Jenkinson et al., 2013*).

## Limitations of the study

Before further discussing the significance of our findings in detail, some of the limitations of this study should be emphasized. First and foremost, the experiments were conducted in patients with Parkinson's disease only; given the invasive character of intracranial recordings, no healthy control group could be tested which limits the generalizability of the reported results. However, as patients performed the task on their regular medication, the physiological basal ganglia activity was restored to the greatest possible extent. Previous studies that report similar movement-induced oscillatory changes across several nuclei of the basal ganglia in patients with different pathologies such as tremor or dystonia (*Androulidakis et al., 2007*; *Brücke et al., 2012*, *2013*) and healthy non-human primates (*Courtemanche et al., 2003*; *Connolly et al., 2015*) further suggest generalizability to the physiological brain. While the study was designed to explore changes in gamma activity with movement amplitude and velocity, grip strength was not monitored. Therefore, varying grip forces across

trials are may have added to the association of movement velocity and gamma activity. However, rotation of the handle required a coordinated movement of proximal and distal arm muscles. Maximal movement velocity correlated across trials and patients with gamma activity, which is unlikely related to sole changes in grip strength. Regarding the comparison to neuronal activation patterns in the OFF medication condition, it has to be considered that recordings shortly after deep brain stimulation surgery may be confounded by a transitional symptom alleviation due to electrode implantation and consecutive edema, the so-called stun effect (*Chen et al., 2006*; *Mann et al., 2009*). Indeed, 4 out of 11 patients recorded OFF medication were not included in the direct ON/OFF comparison, as they showed no relevant deterioration in symptom severity. Another inherent limitation is the lack of histological verification of correct electrode position inside the subthalamic nucleus in deep brain stimulation patients. Nevertheless, electrode placement was guided by intraoperative microelectrode recordings and controlled postoperatively using the Lead-DBS toolbox (*Horn and Kühn, 2015*). Moreover, effective intraoperative macrostimulation and a mean improvement of ~54% in motor symptoms (as assessed by UPDRS-III score) during chronic deep brain stimulation provide further evidence for correct deep brain stimulation lead location. Bipolar recordings of adjacent contact pairs additionally assure the focal origin of the analyzed local field potentials by avoiding volume conduction from more distant areas. The recordings from DBS-macro electrodes did not allow assessment of multi-unit activity. The overlap of changes in gamma power with asynchronous bursts of local spiking activity thus cannot be addressed conclusively. Indeed, broad gamma synchronization spreading between 30 and 200 Hz can be contaminated by parallel increases in neuronal firing rate (*Pesaran et al., 2002*; *Mukamel et al., 2005*; *Rasch et al., 2008*). However, the narrowband gamma peaks in averaged power spectra during movement and the spectrally distinct (40–90 Hz) correlations to maximal velocity hint toward modulations in rhythmic activity instead of an uncoordinated increase in local neuronal firing (*Fries et al., 2007*; *Manning et al., 2009*; *Buzsáki and Wang, 2012*).

## Movement-related gamma synchronization reflects movement velocity or motor motivation

The basal ganglia are suggested to play a pivotal role in both movement initiation and execution. Because gamma synchrony of cortical and subcortical structures accompanies movement rather than preceding it (*Androulidakis et al., 2007*; *Brücke et al., 2012*), it seems less related to motor planning than to monitoring or execution of ongoing movement without primarily depending on somatosensory feedback (*Fischer et al., 2017*). Velocity-dependent scaling of gamma band activity has been interpreted as physiological motor function in the internal globus pallidus (*Brücke et al., 2012*), the major efferent target of the subthalamic nucleus. Interestingly, subthalamic efferences have recently been shown to entrain the activity in downstream structures (*Deffains et al., 2016*) and gamma synchrony seems to play an important role in such network communication(*Fries et al., 2007*). The motor circuit is coupled in a narrow gamma band while movement is executed, a coupling that is strengthened by levodopa and shows an inverse correlation with symptom manifestation (*Litvak et al., 2012*). Thus, movement execution is likely dependent on gamma-band synchronization in local and distant basal ganglia-cortex motor networks and its disturbances may lead to reduced motor vigor in Parkinson's disease. Motor vigor describes the gain of the parameters for a purposive action, that is, speed or amplitude (*Turner and Desmurget, 2010*; *Panigrahi et al., 2015*; *Tan et al., 2015*) and increasing evidence from animal models suggests that the encoding of motor vigor, with special emphasis on movement speed, is strongly dopamine dependent (*Panigrahi et al., 2015*; *Yttri and Dudman, 2016*). Several studies have shown in patients with Parkinson's disease ON medication that subthalamic gamma synchrony scales with kinematic parameters albeit primarily focusing on force instead of velocity modulated movements (*Anzak et al., 2012*; *Joundi et al., 2012*; *Tan et al., 2013*; *Tan et al., 2016*; *Fischer et al., 2017*). Given that patients with Parkinson's disease OFF medication display increasingly bradykinetic and decrementing movements as signs of reduced motor vigor, the question arises whether dopamine affects the scaling capacity within the gamma frequency range. Indeed, ON-OFF medication comparisons in this regard have so far been missing, especially regarding movements that the patients are actually impaired in performing in the dopamine-depleted state. In the present study, the monitoring through gamma synchrony scaling turned out to be more precise ON than OFF medication as the general power decrease OFF

medication was accompanied by a less discriminative modulation of gamma power that may have contributed to the slowing in peak-trial velocity in large and fast movements.

## Gamma synchronization occurs in bursts that reduce in rate in the dopamine-depleted state

In the basal ganglia, synchronization in the beta band has been shown to primarily rely on transient bursting activity that increases in rate during striatal post-movement beta synchronization in task-time-specific patterns (*Feingold et al., 2015*). In the parkinsonian resting state, beta burst duration seems to be pathologically prolonged (*Tinkhauser et al., 2017*). So far, subcortical gamma synchronization as seen during motor performance has not been investigated in this regard while cortical gamma bursts seem to vary remarkably in rate during memory processing which has been attributed to a selective routing potential near the onset of oscillatory synchrony (*Lundqvist et al., 2016*; *Palmigiano et al., 2017*). Here, we provide first evidence for discrete oscillatory dynamics in the gamma frequency range of the subthalamic nucleus during motor processing that showed increasing burst amplitude, duration and rate compared to baseline activity with the gamma burst rate correlating best with averaged increases of gamma power and showing similar scaling with movement velocity. Most interestingly, the movement-related bursting rate correlated with the patients' clinical state and was significantly reduced in the dopamine-depleted state while burst amplitude and duration remained stable in both medication states. As complementary gamma and beta burst occurrence has been described in working memory tasks (*Lundqvist et al., 2016*), one could hypothesize that the afore evoked disadvantageous distribution toward long beta bursts in Parkinson's disease (*Tinkhauser et al., 2017*) would be more pronounced in the dopamine-depleted state thereby impeding fast repetitive synchronization in the gamma band that is needed for increasingly effortful movements. This could, however, not be tested here, as beta burst duration in Parkinson's disease has been reported to rise above 1000 ms and performed movements did not exceed 600 ms.

## A spatially confined gamma oscillator in the dorsolateral subthalamic nucleus

The basal ganglia are functionally segregated into parallel cortico – subcortical loops (*Alexander and Crutcher, 1990*) that result in partially overlapping segments of structural connectivity. We have recently reproduced this segregation on the level of resting oscillatory activity by demonstrating that the dorsolateral part of the subthalamic nucleus, most connected to cortical motor regions, was dominated by beta resting activity (*Accolla et al., 2016*), whereas higher resting alpha power was associated with a more anterior recording location in PD patients (*Horn et al., 2017b*). This study extends these findings to the precise localization of movement-related gamma synchronization. The significant correlation between contact pair location and movement-related gamma power indicates a spatially confined oscillator within the dorsolateral part of the subthalamic nucleus. Extracellular recordings of subthalamic neurons have previously characterized a somatotopic organization of the human subthalamic nucleus that showed higher local activity with arm movements localized more laterally than leg-related neurons (*Rodriguez-Oroz et al., 2001*). Given that subjects in this study performed rotatory arm movements, the significant increase of gamma power toward more lateral electrode positions may relate to the somatotopic organization of the subthalamic nucleus. Interestingly, the percentage of units oscillating at gamma frequency correlates negatively with the bradykinesia scores (*Sharott et al., 2014*) which additionally points toward a link between gamma oscillations and multi-unit activity as interconnected prokinetic features in the subthalamic network.

## Conclusions

Disease-specific oscillations have reliably and repeatedly been associated to symptom severity – most prominently regarding the level of subthalamic resting beta synchrony in Parkinson's disease (*Silberstein et al., 2003*; *Kühn et al., 2006*; *Neumann et al., 2016b*; *Kühn and Volkmann, 2017*). Its potential role in motor slowing is further supported by direct beta frequency stimulation effects at different nodes of the motor network leading to increased bradykinesia in Parkinson's disease patients (*Fogelson et al., 2005*; *Chen et al., 2007*; *Eusebio et al., 2008*). In contrast, synchronization in the gamma band using motor cortical transcranial alternating current stimulation speeds up

movement in healthy subjects (*Joundi et al., 2012*) and deep brain stimulation at individual gamma band frequency improves motor symptoms in Parkinson's disease (*Tsang et al., 2012*). Our results provide additional evidence for a pathophysiological link between reduced movement-related gamma activity and Parkinson's disease by showing a negative correlation between gamma synchronization during movement and motor impairment across 16 Parkinson's disease patients. Moreover, our findings hint toward a potential marker for adaptive deep brain stimulation, considering that we were able to show that movement-related gamma synchronization indicates the current motor state in a spatially confined area while other disease-specific patterns such as the beta peak attenuates significantly during movement. Indeed, subthalamic activity in the gamma band has recently been used to decode the temporal profile of gripping force (*Tan et al., 2016*), which further supports the notion of gamma activity comprising detailed information about specific aspects of movement performance rather than just encoding motor effort. This hypothesis is further supported by our finding that subthalamic gamma synchronization occurs in oscillatory bursts and seems to convey information by varying the rate of gamma bursts instead of their mere amplitude or duration which has been proven to be especially efficient in organizing information in distant networks (*Lundqvist et al., 2016*; *Palmigiano et al., 2017*). The exact pattern of these discrete dynamics and their coding capacity within specific motor programs should be investigated in future studies.

## Methods and materials

### Patients and surgery

Sixteen patients with idiopathic Parkinson's disease (mean disease duration 11.8 years, range 4–20 years; mean age 59.5 years, range 39–75 years; four women; further clinical details given in *Table 1*) took part in this study. All participants provided written informed consent which was approved by the local review boards of the Charité- Universitätsmedizin Berlin and Hannover Medical School and

**Table 1.** Clinical details

| Case | Age/ Gender | Surgical center | Disease duration (Years) | Predominant symptoms | UPDRS med ON/OFF | Δ UPDRS [%] Stim ON/OFF* | Medication decrease* |
|---|---|---|---|---|---|---|---|
| 1 | 59/M | Berlin | 10 | Bradykinesia, Rigidity | –/26 | 42% | NA |
| 2 | 44/M | Berlin | 12 | Bradykinesia, Rigidity, Resting Tremor | 31/41 | 25% | NA |
| 3 | 42/M | Berlin | 4 | Action Tremor, Resting Tremor, Gait Disturbance | 16/42 | 60% | 100% |
| 4 | 53/M | Berlin | 8 | Bradykinesia, Rigidity, Resting Tremor | 3/20 | 53% | 68% |
| 5 | 70/M | Berlin | 19 | Bradykinesia, Rigidity, Resting Tremor | 38/48 | NA | NA |
| 6 | 55/M | Berlin | 15 | Peak-Dose Dyskinesia, Wearing-Off | 18/32 | 75% | 50% |
| 7 | 71/F | Berlin | 18 | Bradykinesia | 17/34 | 39% | 45.5% |
| 8 | 68/M | Berlin | 20 | Speech Difficulties, Hypokinesia | 16/20 | 40% | 61.3% |
| 9 | 67/M | Berlin | 14 | Bradykinesia | 14/24 | 74% | 50% |
| 10 | 49/M | Hannover | 7 | Rigidity | 19/40 | 50% | 14.3% |
| 11 | 39/F | Berlin | 6 | Resting Tremor, Action Tremor | 21/– | NA | 68.4% |
| 12 | 71/M | Berlin | 4 | Resting Tremor | 20/23 | NA | 55.2% |
| 13 | 68/F | Berlin | 5 | Resting Tremor | 14/20 | NA | 74.3% |
| 14 | 58/M | Berlin | 12 | Bradykinesia, Fluctuations, Freezing | –/35 | 69% | 36.3% |
| 15 | 75/M | Berlin | 15 | Action Tremor, Resting Tremor | 20/38 | 69% | 50% |
| 16 | 53/M | Hannover | 10 | Bradykinesia, Rigidity | 24/42 | 53% | 33.3%% |

*after 3 months post-operatively ** NA = not available archive data.
DOI: https://doi.org/10.7554/eLife.31895.014

in accordance with the standards set by the Declaration of Helsinki. All patients underwent stereotactic functional neurosurgery for bilateral implantation of deep brain stimulation electrodes in the subthalamic nucleus. The surgical procedure has been described previously (*Kühn et al., 2005*). Deep brain stimulation electrode extension cables were externalized for 1–7 days, offering access to subthalamic recording sites. The permanent quadripolar macroelectrode used was model 3389 (Medtronic Neurological Division, Minneapolis, MN, USA) with four platinum-iridium cylindrical surfaces (1.27 mm diameter, 1.5 mm length) and a center-to-center separation of 2 mm. Its contacts are numbered 0, 1, 2, and 3, with 0 being the most caudal and 3 the most cranial contact. The intended coordinates at the tip of the electrode (contact 0) were 12 mm from the midline, 0–4 mm behind the midcommissural point and 4–5 mm below the anterior-posterior commissure determined by T2-weighted magnetic resonance images adjusted to the individual patient's anatomy. Electrode placement was refined by microelectrode recordings displaying typical activity patterns once the dorsal border of the subthalamic nucleus was reached (*Hutchison et al., 1998*), effective intra-operative macro-stimulation and postoperative stereotactic imaging. In all cases except of case 10 and 16, the electrode position was additionally controlled using the Lead-DBS toolbox (*Horn and Kühn, 2015*) based on post-operative imaging. Specific methodology is described elsewhere (*Horn and Kühn, 2015*; *Horn et al., 2017a*, *2017b*). Briefly, postoperative acquisitions were linearly coregistered to preoperative acquisitions and normalized to standard stereotactic (ICBM 2009b NLIN Asym; 'MNI') space using DARTEL (*Ashburner, 2007*) as implemented in the Statistical Parametric Mapping toolbox (SPM12; http://www.fil.ion.ucl.ac.uk/spm/software/spm12/). Contacts can then be related to a version of the Morel atlas (*Morel, 2013*) that was likewise defined in MNI space (*Jakab et al., 2012*). All patients had at least one contact within the subthalamic nucleus based on postoperative imaging (except for the right electrode in Case 1). Specifically, 69/84 contact pairs (82%) of 14 patients had at least one contact inside the subthalamic nucleus. The remaining 15 contact pairs from 7 patients either lay below the edge of the target (Case 6: L01), dorso-lateral to the subthalamic border (Case 2: R23, L23; Case 8: R23, L23; Case 9: R23, L23; Case 13: R23, L23; Case 14: R23, L12, L23) or outside the subthalamic nucleus (Case 1: R01, R12, R23). Contact pairs outside the subthalamic nucleus (n = 15) were excluded from further analysis. For an overview of all electrode localizations and a single example of a reconstructed electrode pair position, see *Figure 7—figure supplement 1*.

## Experimental paradigm

Local field potentials were recorded while patients engaged in a reaction time task in which they were asked to perform forearm pronation movements of 3 amplitudes (30°, 60° and 90°) in response to imperative visual cues. They were comfortably seated in front of a 15-inch laptop screen and held a light, rotatable grip with an integrated potentiometer that was able to sense rotatory movement. The investigator visually controlled for a steady posture and unchanged holding of the rotatory device throughout the recording session. On screen, the current handle position was shown as a red dot, the requested angular route as a black semicircle that changed to blue once the patient reached the target position. After 3 s, the color changed back to black, indicating that the patient should return to the starting position. The cues, labeled as 'small', 'medium' and 'large', were presented in randomized order and with inter-trial periods of 4 s (see *Figure 1*). Subjects were instructed to perform the task as quickly and accurately as possible. Before the LFP-recording a brief training session of 5–10 trials was performed. For each arm, patients performed at least 45 trials (15 per condition) during the recording session. The amplitude and duration of the presented cues and the consecutive rotation movements were recorded in parallel to local field potential signals via a 1401 DA converter using Spike 2 software. It has previously been shown in patients with idiopathic isolated dystonia that the three executed movement amplitudes also reflect different movement speed entities (*Brücke et al., 2012*), which we verified for the tested Parkinson's disease patient cohort.

## Recordings

Recordings were made 1–4 days postoperatively while the deep brain stimulation electrodes were externalized. All patients performed experiments in 30–60 min following the intake of their regular dopaminergic medication (ON medication state). In 11/16 patients, experiments were also performed after they had been at least 12 hours withdrawn from dopaminergic medication. Patients

were considered OFF medication when they showed ≥30% UPDRS III score increase compared to UPDRS III score under dopaminergic substitution. 4 patients did not meet this criterion despite medication withdrawal, most likely due to the post-operative stun effect and were therefore not considered for further comparisons between the ON and OFF state. 10 patients performed the paradigm with both hands consecutively, four patients only with their left and two patients only with their right hand. In the OFF-medication state, an additional 2 patients performed it only with the left hand due to severe tremor or rigidity that obviated test performance in those patients. This led to a total of 26 sides from 16 patients in the ON-medication and 11 sides from 7 patients in the OFF-medication state that were finally analyzed. From the 7 patients included in the ON-OFF comparison, three performed the task first ON then OFF medication and 4 patients the other way around to limit order effect. Local field potentials were recorded in a bipolar montage between adjacent contact pairs (0–1, 1–2, 2–3). Signals were band-pass filtered between 1 and 250 Hz, amplified (x 50.000) using a D360 amplifier (Digitimer Ltd, Welwyn Garden City, Hertfordshire, UK), and sampled at 1 kHz in parallel with the movement traces using a 1401 AD converter.

## Data analysis

Analyses of both behavioral and electrophysiological data were performed in MATLAB (version R2016a; The MathWorks, Natick) using custom MATLAB code based on the Statistical Parametric Mapping (*Litvak et al., 2011b*Litvak et al., 2011) (SPM) and Fieldtrip (*Oostenveld et al., 2011*) toolboxes (https://github.com/RoxanneLofredi/stn). Continuous recordings were divided into epochs of 5 s (−2 to +3 s around movement onset of each trial). Fifth order butterworth, high-pass (>1 Hz) and notch filters (48–52 Hz) were applied to limit effects of line noise. Epochs were rejected from further analysis if they contained artifacts and saturation in the local field potentials trace or abnormalities in the movement trace, leaving a mean (±SEM) of 27.6 ± 10.8 remaining trials in the ON-medication state and 27.0 ± 9.0 (mean ±SEM) in the OFF-medication state per hand and per condition for final analysis. The time points of the maximal amplitude (maxA), maximal velocity (maxV) and movement onset (MO) of each trial were automatically detected on the basis of the movement trace. Maximal amplitude was set as the highest point in the movement trace per trial. The velocity trace was defined as the first derivative of the movement trace and its highest point as maximal velocity. Movement onset was automatically defined as the time point where 10% of the peak-trial velocity were reached. All automatically defined time points were visually checked and adjusted if necessary. Reaction time was calculated as the interval between the imperative visual cue and movement onset. Accuracy was defined as the percentage deviation of a reference scaling movement, performed before starting the task. To investigate the changes in local field potential activity in the time-frequency domain, a time-frequency decomposition based on Morlet wavelets with seven cycles was applied to the local field potential recordings of all contact pairs from each trial. Local field potentials were analyzed over frequencies between 1 and 100 Hz with a frequency resolution of 1 Hz. Event-related local field potential power change was subsequently normalized in percentage relative to an averaged pre-trial baseline period (−2 to 0.5 s preceding the visual cue), smoothed over 6 Hz and 200 ms and averaged using robust averaging across trials and contact of pairs of the same patient. Normalized time frequency plots of each movement condition (small, medium, large) were aligned to movement onset, maximal velocity and maximal amplitude, averaged across contacts, hemispheres (ipsi- and contralateral) and patients for visualization. If not otherwise indicated, analyses were separately performed for both subthalamic nuclei ipsi- and contralateral to the moved hand.

## Selection of time-frequency bands

For the main analysis, distinct frequency ranges and a time period of interest were designated. Frequency bands were determined according to significant cluster of movement-related modulation in the grand average from all patients during movement (0–500 ms after movement onset; theta-band 2–8 Hz and beta-band 13–30 Hz). As the study focuses on velocity-related modulation of gamma activity, the gamma band was restricted to the frequency range that showed modulation between conditions (gamma-band 40–90 Hz), see *Figure 2—figure supplement 1*. In order to compare varying movement characteristics, mean power changes were averaged around a time period of 500 ms following movement onset, which included significant differences in amplitude and velocity as seen

in the behavioral results while assuring an ongoing movement in all conditions. Thus, if not indicated otherwise, power changes averaged over time and frequency refer to the mean power changes in the theta, beta or gamma band from time points 0 to 0.5 s after movement onset.

## Relative power changes in averaged time-frequency plots

We examined whether power changes averaged across all patients differed from baseline during movement, and compared time-frequency bins between the 3 task conditions (small, medium, large). We then tested if movement-related oscillatory patterns differed between the ipsi- and contralateral subthalamic nucleus to the moved hand by comparing averaged relative power changes in the pre-defined theta, beta and gamma bands. Mean relative power changes as seen when aligned on movement onset were juxtaposed to an alignment on the time points maximal velocity and maximal amplitude, in order to identify the condition that reflected best the maximal relative power change. Then, mean relative power changes in the 3 predefined time-frequency ranges were compared between conditions to determine an eventual parametric modulation of event-related changes in oscillatory activity.

## Correlation between event-related power changes and behavioral parameters and clinical state

To evaluate the temporal and spectral specificity of movement-related oscillatory activity, the within-patient correlation between maximal inter-trial velocity and power changes in each time-frequency bin was calculated on a single-trial level (Spearman's $\rho$) and results were tested for significance using one sample tests. In the same way, oscillatory activity was also correlated with single-trial amplitude, accuracy and reaction time. On a group level, correlation coefficients of the entire power spectra from all patients were tested against the null hypothesis using permutation tests and False Discovery Rate (FDR) correction. Relative power changes in the subthalamic nucleus contralateral to the moved side were averaged across conditions for each of the three frequency bands of interest and correlated across all patients with the UPDRS-ON score using non-parametric Spearman rank correlation.

## Dopamine-dependence of movement-related modulation in oscillatory patterns

In a subpopulation of 7 patients, the influence of dopamine on movement-related modulation of oscillatory activity was assessed by matched group comparison of the mean relative power changes for the theta, beta and gamma time-frequency bands with and without dopaminergic medication. For visualization, averaged time-frequency plots in the ON- and OFF- medication state were shown separately for this subgroup. Only results of the subthalamic nucleus that was contralateral to the moved hand are shown.

## Burst determination and analysis

For burst determination, the criteria that have previously been established by *Tinkhauser et al. (2017)* for beta burst analysis in patients with Parkinson's disease were applied. The raw local field potential recording was band pass filtered (Fifth order butteworth filter) around the gamma band (40–90 Hz), rectified and smoothed with moving average gaussian smoothing kernel of 100 ms length. A gamma burst was determined when the resulting instantaneous power exceeded the 75th percentile of the signal amplitude distribution of all data in each electrode. Threshold crossings lasting shorter than one gamma cycle were excluded from the analysis (see *Figure 7A*). For ON-OFF comparisons (n = 7), the threshold of the ON medication state was applied to the OFF recordings of the matching hemisphere. The amplitude of a gamma burst was defined as the area under the curve between signal and threshold line. The density of gamma bursts, in the following 'burst rate', was defined as the number of bursts occurring in 0.5 s. The change of burst rate therefore serves as an estimate of the probability of bursting during movement (0.5 s following movement onset) compared to that in 0.5 s of baseline activity (2–1.5 s before movement onset). We also compared averaged burst duration and amplitude during movement with that during baseline activity. In a second step, we investigated which movement-related burst property changes (amplitude, duration or rate) correlated most with the movement-related increase of power in the gamma band. Correlation coefficients between power changes and changes in burst properties were calculated for each bipolar

recording, and mean values from contact pairs of the same electrode (n = 26) were averaged on the group level and tested for differences using permutation tests. For the property that correlated best, we investigated whether it also exhibited scaling with task condition (small, medium, large) by averaging across hemispheres and patients (n = 16) as we did for averaged gamma power. Additionally, its relative change averaged across conditions was correlated with the clinical state as assessed by the UPDRS-III ON score, available in 14 patients ON medication. In the 7 patients that performed the task both ON and OFF dopaminergic medication, gamma burst rate, amplitude and duration were averaged across contact pairs and ON-OFF comparison was performed separately for each hemisphere (n = 10) using permutation testing.

## Spatial distribution of averaged power change in the gamma band

To create a spatial map of oscillatory changes in normalized three-dimensional anatomical space, all contact pair MNI coordinates from electrodes contralateral to the moved hand were extracted from electrode localization results following the subcortical electrophysiology mapping approach described in *Horn et al. (2017b)*. Briefly, contact pair coordinates from the left hemisphere were flipped to the right hemisphere, averaged gamma power (see above) was then assigned to the respective contact pairs for each movement condition (small, medium, large), aligned to the time point of maximum velocity and visualized as a scattered point cloud in MNI space. A scattered interpolant was fit to these data points and values were projected to an equidistant grid using natural neighbor interpolation. This resulted in a volumetric description of gamma power distribution, which was then projected to the surface of the subthalamic nucleus using Surf Ice (Open Source; https://www.nitrc.org/projects/surfice/). In addition, a binary mask of the subthalamic nucleus motor part (*Ewert et al., 2017*) was projected onto the surface of the subthalamic nucleus for visual comparison. Non-parametric rank-based Spearman correlations were calculated between averaged gamma power across all movement conditions and respective contact-pair locations on the X, Y, Z axes to test for a gamma gradient in MNI standard space. This was done to show a potential group effect of a statistical relation between recording site and movement-related gamma synchronization. Thereby potential group effects of contact pair location on movement-related gamma synchronization are verified, remaining unaffected by the described masking and interpolation.

## Statistical analysis

Non-parametric Monte Carlo permutation tests were used for statistical results reported in this study. Permutation tests do not rely on assumptions about the underlying data distribution. In brief, the sample of tested values was interchanged randomly 5000 times to generate a probability distribution in which the observed original sample rank is reported as the p-value. All tests were multiple comparisons corrected by controlling the false discovery rate for an α-level of α = 0.05. Only p-values smaller than the false discovery rate-corrected threshold were considered significant. When investigating power changes, each time-frequency bin was tested separately and only significant p-values that extended over a cluster of at least 150 time-frequency bins are shown. Rank-based Spearman correlations were calculated if data deviated significantly from a normal distribution as assessed by Kolmogorov-Smirnov tests. Otherwise, linear Pearson correlations were conducted.

# Acknowledgements
We would like to thank all patients for their participation.

# Additional information

## Funding

| Funder | Grant reference number | Author |
| --- | --- | --- |
| Deutsche Forschungsgemeinschaft | DFG KFO247 | Andrea A Kühn |

The funders had no role in study design, data collection and interpretation, or the decision to submit the work for publication.

## Author contributions

Roxanne Lofredi, Conceptualization, Resources, Data curation, Software, Formal analysis, Investigation, Visualization, Methodology, Writing—original draft, Writing—review and editing; Wolf-Julian Neumann, Conceptualization, Software, Formal analysis, Supervision, Visualization, Methodology, Project administration, Writing—review and editing; Antje Bock, Investigation, Methodology, Writing—review and editing; Andreas Horn, Software, Formal analysis, Methodology, Writing—review and editing; Julius Huebl, Sandy Siegert, Investigation, Writing—review and editing; Gerd-Helge Schneider, Resources, Data curation, Funding acquisition, Investigation, Project administration, Writing—review and editing; Joachim K Krauss, Resources, Data curation, Investigation, Writing—review and editing; Andrea A Kühn, Conceptualization, Resources, Data curation, Supervision, Funding acquisition, Investigation, Methodology, Project administration, Writing—review and editing

## Author ORCIDs

Roxanne Lofredi http://orcid.org/0000-0002-1845-8250
Wolf-Julian Neumann https://orcid.org/0000-0002-6758-9708

## Ethics

Human subjects: All participants provided written informed conset which was approved by the local review boeards of the Charité - Universitätsmedizin Berlin and Hannover Medical School and in accordance with the standards set by the Declaration of Helsinki (EA2/071/08).

## Decision letter and Author response

Decision letter https://doi.org/10.7554/eLife.31895.018
Author response https://doi.org/10.7554/eLife.31895.019

# Additional files

## Supplementary files

• Transparent reporting form
DOI: https://doi.org/10.7554/eLife.31895.015

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
