## [Decision Letter]

Thank you for submitting your article "Dopamine-dependent scaling of subthalamic gamma bursts with movement velocity in patients with Parkinson's disease" for consideration by *eLife*. Your article has been reviewed by two peer reviewers, and the evaluation has been overseen by a Senior Editor and a Reviewing Editor. The reviewers have opted to remain anonymous.

The reviewers have discussed the reviews with one another and the Reviewing Editor has drafted this decision to help you prepare a revised submission.

This paper presents a very detailed analysis of changes in oscillatory activity in the subthalamic nucleus during a movement task in which the amplitude and velocity of movements were systematically varied. While STN oscillatory activity during movement and rest has been studied previously, there are some very interesting novel features of this study: a movement task and analytic approach designed to reveal how oscillatory activity scales with movement velocity; the demonstration that movement related gamma changes are actually composed of trial-by-trial changes in brief bursts of gamma activity, and the finding that movement related changes in bursts of gamma activity scale with movement velocity, are affected by dopaminergic state, and accordingly, the disease severity is negatively correlated with the level of gamma burst. This work is very timely in that it follows recent influential work of the Brown group on beta bursts, showing more or less opposite behavior of beta bursts. The paper makes an argument that PD-related changes in gamma bursting may be key in the expression of bradykinesia. In general, the figures and Results section are quite clear given the complexity of the analysis. The reviewers do have however, some concerns.

General Concerns:

1) The gamma bursts are so short that they are on average < 2 cycles of the gamma oscillation, raising the question of whether one can define a gamma oscillation over such a short duration. Is it possible that these gamma bursts reflect underlying bursts of spiking activity, given the short durations of the bursts described here, and the similar duration of spike bursts in STN from human and nonhuman primate micro-recording?

2) Related to the origin of the gamma band activity studied here, the literature on cortical field potentials makes a clear distinction between two types of high frequency phenomena: "broadband gamma" which is asynchronous, spans a wide spectral range approximately between 50 and 200 Hz, and is modulated by movement; versus narrowband gamma phenomena which are associated with a distinct narrow peak in the power spectrum of the field potential, whose exact frequency may vary from 50-90 Hz.

The literature on STN LFP recording is in general much less clear and it is hard to tell from this paper, which type of phenomena is being studied. This distinction should be addressed in the Discussion, along with a consideration of whether the gamma activity studied is actually oscillatory or not. In this paper, the gamma activity is isolated by relatively broad bandpass filtering 40-90 Hz. Some figures (Figure 2, small amplitude) suggest more of a broadband phenomenon while others suggest more narrowband. This distinction is mechanistically important. Power spectra in the on state during rest and movement for individual subjects could be helpful if they show a narrow peak versus a broader band change. It seems very plausible that synchronized bursts of STN spiking could show up in LFP recordings as brief gamma band elevations in the LFP thus the authors should consider this mechanistic possibility.

Further clarification of the choice of definition of the gamma band would be useful. Why has the range 30-40Hz been excluded from the analysis? Perhaps some further annotation of Figure 2—figure supplement 2 would be of help in this regard.

3) Clarifications/Questions:a) Did the subjects undergo a training period prior to the experiment?

b) Were fatigue effects accounted for as the trials progressed? Did this create a discord between encoded effort (gamma burst) and motor output (achieved velocity)? Insights into such a phenomenon may give us a better idea as to exactly what aspect of movement the gamma activity encodes.

c) Were LFP changes related to maintenance of posture accounted for (as a baseline)? In a similar vein, might an increase in grip-strength have gone hand-in-hand with the increased movement velocity (and thus have also been encoded by the gamma burst power?)

4) The Introduction it is quite verbose and while the discussion of the difference between "finely tuned" and "broad" gamma activity is important, as noted above, the treatment of this in the Introduction is unclear and the fine/broad distinction might best be just introduced briefly in the Introduction and dealt with more fully in the Discussion.

5) The Discussion is also quite verbose as well, and requires some revision to address the concerns described above. It states, for example, that movement related changes in beta band were not dopamine dependent. This needs some further discussion as it appears conceptually at odds with prior literature. Since dopamine has a major effect onSTN beta band activity, based on many prior papers, it seems possibly contradictory that there is no effect on movement related changes in beta.

6) Abstract: the use of the word "seemed" in the Abstract is unnecessary.

7) Materials and methods: in selection of time frequency bands – this statement is unclear: "Specific frequency bands were determined as standard frequency bands according to the main effects of event-related oscillatory changes seen in the grand average from all patients" – does this refer to the center frequency of each predetermined band? Determined by the most movement reactive frequencies within each band? Needs clarification.

8) Figure 3 legend is unclear and may be mislabeled. D appears to be related to C, (with significance threshold?).

---

## [Author Response]

General Concerns:1) The gamma bursts are so short that they are on average < 2 cycles of the gamma oscillation, raising the question of whether one can define a gamma oscillation over such a short duration. Is it possible that these gamma bursts reflect underlying bursts of spiking activity, given the short durations of the bursts described here, and the similar duration of spike bursts in STN from human and nonhuman primate micro-recording?

We thank the reviewer for raising this important point. Before addressing it in detail, it might be important to highlight that the LFP recordings displayed a continuous gamma oscillation (see Figure 6) and only the crossing of a predefined amplitude threshold lasted between 1-3 cycles depending on the peak frequency. It can be assumed that peak amplitude fluctuates in every gamma cycle and short elevations over few cycles are not generally in conflict with the definition of an oscillation.

The underlying mechanisms leading to amplitude changes in brain oscillations are an ongoing debate and the exact contribution of single bursts of spiking activity to rhythmic oscillations, especially in higher frequency bands, is difficult to work out. Multi-target recordings from cortical and subcortical structures may help to figure out whether the brief increase of gamma amplitude for 1-2 cycles during movement is traceable at the network level, which would be in favor of its oscillatory nature. Using Granger causality-based analysis of information flow, Litvak et al. (2012) were able to show that cortical gamma oscillations are driven from the subcortical gamma signal during movement and M1-STN coherence is spectrally specific to the 60 – 90 Hz range. Moreover, STN firing during movement could be shown to be synchronized to the envelope of gamma oscillations in primary motor cortex but not local STN with accompanying cell-specific up- or down-regulation of firing rates in PD patients (Lipski et al., 2017). Taken together, these results support the concept of precise timing in gamma power changes across the motor circuit leading to changes in local spiking activity rather than vice versa.

The specific contribution of bursty neurons in the STN to the movement-related gamma synchronization has to our knowledge not been investigated yet. High proportion of subthalamic neurons with burst firing seems a pathological feature of PD that correlates with beta power in in humans (Steigerwald et al., 2008) and increases in the dopamine-depleted state in non-human primates (Hammond et al., 2007). Thus, it is conceptually unlikely that apparently pro-kinetic gamma bursts predominantly reflect such underlying bursts of spiking activity. Ultimately, only multi-unit recordings could quantify the contribution of local bursts of spiking activity to brief transitory increases in movement-related gamma power.

To emphasize that we cannot definitely exclude the contribution of asynchronous bursts of local spiking activity to the brief bursts of increase in gamma amplitude in this study, we made the following changes to the manuscript:

Discussion section

“The recordings from DBS-macro electrodes did not allow assessment of multi-unit activity. […] However, the narrowband gamma peaks in averaged power spectra during movement and the spectrally distinct (40-90 Hz) correlations to maximal velocity hint towards modulations in rhythmic activity instead of an uncoordinated increase in local neuronal firing (Fries et al., 2007; Manning et al., 2009; Buzsaki and Wang, 2012).”

2) Related to the origin of the gamma band activity studied here, the literature on cortical field potentials makes a clear distinction between two types of high frequency phenomena: "broadband gamma" which is asynchronous, spans a wide spectral range approximately between 50 and 200 Hz, and is modulated by movement; versus narrowband gamma phenomena which are associated with a distinct narrow peak in the power spectrum of the field potential, whose exact frequency may vary from 50-90 Hz.The literature on STN LFP recording is in general much less clear and it is hard to tell from this paper, which type of phenomena is being studied. This distinction should be addressed in the Discussion, along with a consideration of whether the gamma activity studied is actually oscillatory or not. In this paper, the gamma activity is isolated by relatively broad bandpass filtering 40-90 Hz. Some figures (Figure 2, small amplitude) suggest more of a broadband phenomenon while others suggest more narrowband. This distinction is mechanistically important. Power spectra in the on state during rest and movement for individual subjects could be helpful if they show a narrow peak versus a broader band change. It seems very plausible that synchronized bursts of STN spiking could show up in LFP recordings as brief gamma band elevations in the LFP thus the authors should consider this mechanistic possibility.Further clarification of the choice of definition of the gamma band would be useful. Why has the range 30-40Hz been excluded from the analysis? Perhaps some further annotation of Figure 2—figure supplement 2 would be of help in this regard.

We thank the referees for bringing to our attention that it was not made sufficiently clear which high frequency phenomenon we are discussing in this publication. Here, we focus on the narrow band gamma activity that was consistently increased with behavioral test conditions. This is in line with previous publications on scaling of narrowband gamma power with kinematic parameters in the basal ganglia (Anzak et al., 2012; Joundi et al., 2012; Tan et al., 2013, 2016; Fischer et al., 2017). However, as the referees correctly pointed out, both narrow- and broadband changes occur in the STN during movement. As shown in the averaged time frequency plots in Figure 2, there is a broad increase in high frequency activity (>40 Hz) during movement. The broadband synchronization may partly relate to asynchronous activity during movement. However, correlation of spectrally specific power scaling with maximal velocity on a single trial level was confined to the narrow gamma band between 40-90 Hz (as shown in Figure 3). Therefore, we hypothesize that two co-occurring phenomena take place during movement: a synchronous oscillation that scales with motor output and an asynchronous broadband activity that may reflect local spiking activity without a direct behavioral correlate.

To visualize that movement-related increases in gamma power occurred in narrow band peaks, we followed the reviewers’ recommendation and added a figure both with averaged power spectra across patients and from a single patient at rest and during movement. In the averaged power spectrum, the frequencies that display a significant change between rest and movement are also labeled (highlighted in grey, see Figure 2—figure supplement 1). This new figure was integrated in the manuscript. The band between 30 – 40 Hz has been excluded because it contains periods of desynchronization at movement onset and synchronization in later phases of the trial and was generally not significant in the grand average time frequency analysis (see also Figure 5 and supplementary Figure 2—figure supplement 1).

To emphasize the differentiation between narrow- and broadband changes in the gamma range we revised the first paragraph of the Discussion that now reads as follows:

“Our results provide first evidence for dopamine-dependent and finely-tuned scaling of movement-related gamma synchronization that occurs in transient bursts and is spatially confined to the dorsolateral part of the subthalamic nucleus in patients with Parkinson’s disease. […] Therefore, we hypothesize that two co-occurring phenomena take place in the subthalamic nucleus during movement: a narrowband gamma oscillation that scales with motor output and an asynchronous broadband activity that may reflect local spiking activity without a direct behavioral correlate.”

3) Clarifications/Questions:a) Did the subjects undergo a training period prior to the experiment?

Yes, the subjects underwent a brief training period of 10 trials prior to the experiments. The following changes were made to the manuscript:

“Before the LFP-recording a brief training session of 5 -10 trials was performed.”

b) Were fatigue effects accounted for as the trials progressed? Did this create a discord between encoded effort (gamma burst) and motor output (achieved velocity)? Insights into such a phenomenon may give us a better idea as to exactly what aspect of movement the gamma activity encodes.

Thank you for pointing out this very interesting aspect. A potential fatigue effect would indirectly assess the controversy whether gamma power scales with behavioral output or its effort. Following the reviewers’ suggestion, we re-analysed our data in this regard. For each moved hand (26 sides ON medication, 10 sides OFF medication), we correlated the change in velocity and the change of averaged gamma power over trials in the course of the experiment. There was no general fatigue effect leading to progressively decreasing movement velocity across patients. In single patients, maximal velocity significantly decreased (ON medication, n=5/16; OFF medication, n= 3/7) or increased (ON medication, n=4/16; OFF medication, n= 2/7) over trials. In all cases, this progressive change of velocity across trials occurred only in one of both hands. One may hypothesize that fatigue may have led to a decreasing peak velocity over trials while gamma power stays stable; as fatigue leads to reduced motor output with stable effort. Or stable peak velocity might be accompanied by a continuous increase in gamma power over trials to compensate for the increasing effort demand for the same motor output. However, there was no significant correlation between changes of both variables over trials (Spearman’s Rho=-0.09, P=0.3). Thus, no systematic fatigue effect could be observed in our dataset. This may be related to the fact that subjects had to perform relatively short movements (500-700 ms) that were followed by rather long inter-trial durations (~4 seconds).

**Author response image 1. respfig1:** Changes in maximal velocity and gamma synchronization over trials do not correlate. Spearman’s correlation was conducted for both maximal velocity and averaged gamma power over trials. No robust correlation was seen between the correlation coefficients of both variables (Rho=-0.09, P=0.3).

We incorporated our additional analyses on a potential fatigue effect in the Results section that now reads as follows:

*“*To control for a potential fatigue effect leading to decreased peak velocity over trials or a subsequent compensatory mechanism in the gamma band, we investigated a potential systematic change of movement velocity or gamma band activity over the course of the experiment. There was no significant correlation between either variables over trials (Spearman’s Rho=-0.09, P=0.3) making a systematic fatigue effect unlikely.”

c) Were LFP changes related to maintenance of posture accounted for (as a baseline)? In a similar vein, might an increase in grip-strength have gone hand-in-hand with the increased movement velocity (and thus have also been encoded by the gamma burst power?)

During the entire experiment, the investigator visually inspected the posture and holding of the rotatory device and instructed the subject to maintain those positions stable. The TF-plots were baseline-corrected on a single trial level, therefore changes of posture between trials should not account for changes in movement-related changes. However, we cannot exclude that within a trial a slight change of posture may have occurred which was not systematically monitored.

This information has now been included in the description of the experimental task (Materials and methods section):

“The investigator visually controlled for a steady posture and unchanged holding of the rotatory device throughout the recording session.”

Regarding the grip strength, it was not measured by the device used to track movement trajectories. Hence, we cannot exclude a potential confounder due to varying grip strength across trials. Given that movement velocity correlated on a single trial level with changes in the gamma frequency band (Figure 3), it seems unlikely that changes in grip strength occurred accordingly. Moreover, gamma power peaked at the time point of maximal velocity (Figure 3/B) and one may assume that grip strength would be highest around movement onset. Finally, the rotatory movement of the handle required much more force increase in more proximal arm muscles than grip force. Nevertheless, to account for the potential limitation of our study, we added the following phrase to the Discussion:

“While the study was designed to explore changes in gamma activity with movement amplitude and velocity, grip strength was not monitored. […] Maximal movement velocity correlated across trials and patients with gamma activity, which is unlikely related to sole changes in grip strength.”

4) The Introduction it is quite verbose and while the discussion of the difference between "finely tuned" and "broad" gamma activity is important, as noted above, the treatment of this in the Introduction is unclear and the fine/broad distinction might best be just introduced briefly in the Introduction and dealt with more fully in the Discussion.

We thank the reviewer for this comment to help make the article more concise. To meet the present request, we shortened the Introduction substantially, by removing two paragraphs, and we rewrote the paragraph on movement-related gamma synchronization and added a brief Introduction to the distinction between broad- and narrowband activity:

“In contrast to pathological activity at rest, less is known about the functional role of physiological movement-related changes of oscillatory patterns in the cortico-basal ganglia motor loop. […] Previous studies have shown that narrowband increases in gamma oscillations with a center frequency between 30-90 Hz and ~20 Hz width were of functional significance in encoding of movement parameters (Jenkinson et al., 2012; Joundi et al., 2012; Tan et al., 2015).”

5) The Discussion is also quite verbose as well and requires some revision to address the concerns described above. It states, for example, that movement related changes in beta band were not dopamine dependent. This needs some further discussion as it appears conceptually at odds with prior literature. Since dopamine has a major effect onSTN beta band activity, based on many prior papers, it seems possibly contradictory that there is no effect on movement related changes in beta.

We thank the reviewers for the encouragement to discuss the role of beta desynchronization during movement in greater detail. Our statement regarding the lack of dopamine-dependence on beta desynchronization during movement focused on the averaged percentage of beta change around maximal velocity. The difference in the relative beta power (13-30 Hz) decrease in the 500 ms around maximal velocity (ON: – 29.5 ± 19.5%, OFF: – 22.3 ± 19.3%, p=0.062) did not reach significance. This finding may seem at odds with prior literature as beta power is a robust pathophysiological marker for Parkinson’s disease and is significantly increased in the dopamine-depleted state at rest (Kühn et al., 2006, 2009). It has to be considered that absolute beta suppression with dopaminergic therapy is well established but less is known about the influence of dopamine on relative beta dynamics during movement. Indeed, the few reports mainly focused on relative changes during movement initiation (Doyle et al., 2005; Foffani et al., 2005; Androulidakis et al., 2007; Anzak et al., 2012). We therefore re-analyzed our dataset by averaging the relative change in beta power around movement onset. In line with previous findings, there was significantly stronger beta desynchronization ON than OFF medication (ON: -29.37 ± 19.27%, OFF: -16.36 ± 16.01%, p=0.037, see figure below). This is an interesting aspect, which may relate to the difficulties in movement initiation in PD patients. Furthermore, the results are in line with the hypothesis that beta desynchronization acts as a binary gating function, which allows task-related processing to occur (Tan et al., 2013). To visualize the differential effects of dopamine over time we added a supplementary figure (Figure 2—figure supplement 3) to the manuscript.

The following remarks were added to the manuscript:

Results section:

“Relative changes in beta desynchronization differed between time points. While beta ERD was significantly more pronounced in the ON medication state when averaged around movement onset (ON: -29.37 ± 19.27%, OFF: -16.36 ± 16.01%, p=0.037), this difference did not reach significance during maximal velocity (ON: – 29.5 ± 19.5%, OFF: – 22.3 ± 19.3%, p=0.062), see Figure 2—figure supplement 3.”

Discussion section:

“Movement-related changes in the theta and beta band showed no velocity-related modulation and were similarly pronounced bilaterally. The relative power change around maximal velocity was not dopamine dependent. However, beta desynchronization was significantly higher in the ON vs. OFF condition, when trials were aligned to movement onset, which may indicate an association of beta decrease with movement initiation.”

6) Abstract: the use of the word "seemed" in the Abstract is unnecessary.

We have changed the Abstract accordingly and the phrase now reads as follows:

“These effects relied on movement-related bursts of transient synchrony in the gamma band.”

7) Materials and methods: in selection of time frequency bands – this statement is unclear: "Specific frequency bands were determined as standard frequency bands according to the main effects of event-related oscillatory changes seen in the grand average from all patients" – does this refer to the center frequency of each predetermined band? Determined by the most movement reactive frequencies within each band? Needs clarification.

We thank the reviewers for indicating this imprecision in the Materials and methods section. We chose the frequency bands according to the extent of the significant cluster seen in the grand average from all patients at the time point of movement onset. For the gamma frequency band, we focused on the frequency range that displayed velocity-related modulation as seen in Figure 2 and Figure 3. To clarify the basis of our decision we revised the paragraph in the Materials and methods section:

“Frequency bands were determined according to significant cluster of movement-related modulation in the grand average from all patients during movement (0-500 ms after movement onset; theta-band 2–8 Hz and beta-band 13-30 Hz). As the study focuses on velocity-related modulation of gamma activity, the gamma band was restricted to the frequency range that showed modulation between conditions (gamma-band 40-90 Hz), see Figure 2—figure supplement 1.”

8) Figure 3 legend is unclear and may be mislabeled. D appears to be related to C, (with significance threshold?).

We excuse for the mislabeling of the legend corresponding to Figure 3 and thank the reviewers for noting this error. The section C and D of the legend for Figure 3 has been corrected and now reads as follows:

“(C) Single trial Spearman’s correlations with movement velocity across all time frequency bins aligned to maximum movement velocity (time 0 s) were conducted in all patients. […] No significant time frequency bins were revealed for the ipsilateral hemisphere (data not shown).”